# *Ortho*-Functionalized Dibenzhydryl Substituents in *α*-Diimine Pd Catalyzed Ethylene Polymerization and Copolymerization

**DOI:** 10.3390/polym12112509

**Published:** 2020-10-28

**Authors:** Qasim Muhammad, Wenmin Pang, Fuzhou Wang, Chen Tan

**Affiliations:** 1Department of Polymer Science and Engineering, University of Science and Technology of China, Anhui, Hefei 230601, China; qasimchemist@mail.ustc.edu.cn (Q.M.); pangwm@ustc.edu.cn (W.P.); 2Key Laboratory of Structure and Functional Regulation of Hybrid Materials of Ministry of Education, Institutes of Physical Science and Information Technology, Anhui University, Anhui, Hefei 230601, China

**Keywords:** olefin polymerization, α-diimine, polar monomer, palladium, dibenzhydryl

## Abstract

Sterically bulky diarylmethyl-based ligands have received increasing attention in the field of late-transition-metal catalyzed olefin polymerization. *Ortho*-substituents may have a significant impact on the performance of diarylmethyl-based *α*-diimine Pd catalysts. In this contribution, a series of *α*-diimine Pd catalysts bearing *ortho*-methoxyl/hydroxyl functionalized dibenzhydryl units were prepared, characterized, and investigated in ethylene polymerization and copolymerization with methyl acrylate (MA). The catalytic performances were improved by introducing more *ortho*-substituents. The catalysts exhibited good thermal stabilities at high temperatures, producing branched polyethylenes. The catalysts bearing hydroxyl groups possessing intramolecular H-bonding, resulted in slightly higher incorporation ratios of MA unit when compared with the catalysts bearing methoxyl groups.

## 1. Introduction

Polyolefins have been the world’s largest produced synthetic polymers [1]. Late-transition-metal catalysts can mediate olefin copolymerization with polar monomers [2,3,4,5,6,7,8,9,10,11,12,13,14,15,16,17,18,19,20,21], providing a direct and potentially economic way to access functionalized polyolefins with some improved properties as well as some unique functions [22,23,24,25,26,27,28,29,30,31]. In the 1990s, Brookhart et al. reported the studies of α-diimine Pd catalysts for olefin polymerization and copolymerization with polar monomers ([Fig polymers-12-02509-ch001]A) [32,33,34]. These α-diimine Pd catalysts can produce polyolefins with branched even branch-on-branch structures while using ethylene as the only monomer via the “chain-walking” polymerization mechanism [35,36,37,38,39,40], providing a unique approach for modulating the polyolefin microstructures [41]. However, most α-diimine Pd catalysts exhibit poor thermal stability at high temperatures (>60 °C) due to decomposition reactions that are associated with C–H activation of ligand, associative chain transfer, and bisligation of M–H species, which has limited their practical applications [42,43,44,45,46,47,48,49,50]. It was demonstrated that the formation of a rigid conformation can keep the N-aryl bonds from rotation and, consequently, improve the thermal stabilities of *α*-diimine Pd catalysts. For instance, Guan et al. developed the *α*-diimine Pd catalysts containing sterically rigid cyclophane structures ([Fig polymers-12-02509-ch001]B) [51,52]. Wu and Gao et al. developed the steric bulk backbone strategies for preventing bond rotations in N-aryl moieties ([Fig polymers-12-02509-ch001]C) [53,54,55]. These catalysts exhibit much higher thermal stabilities at 80 °C when compared with the classical Brookhart’s Pd catalysts (generally have no activity above 60 °C). Recently, Chen et al. introduced the sterically bulky diarylmethyl moieties on the structures of the *α*-diimine Pd catalysts ([Fig polymers-12-02509-ch001]D). These catalysts bearing four diarylmethyl groups resulted in improved thermal stabilities of catalysts at high temperatures (80–100 °C), producing semi-crystalline products with low branching densities [46,47,48,49,50]. The polyethylene branching densities can be modulated by reducing the number of diarylmethyl groups [56,57]. The introduction of some para-substituents to the diarylmethyl moieties can result in the modulation of polyethylene branching densities via tuning polymerization temperature [58,59]. Jian et al. reported a series of diarylmethyl-based *α*-diimine Pd catalysts that contain bulky axial pentiptycenyl substituents and a bulky dibenzobarrelene backbone ([Fig polymers-12-02509-ch001]E), producing ultrahigh methyl branching polyethylenes at a wide range of temperature (0–130 °C) [60,61,62].

Many efforts have been directed towards modifying structures of ligands by installing various substituents in order to improve the performances of *α*-diimine Pd catalysts. However, to the best of our knowledge, so far, no α-diimine Pd catalysts bearing *ortho*-functionalized dibenzhydryl substituents have been reported. In this contribution, a series of α-diimine Pd catalysts bearing *ortho*-methoxyl/hydroxyl functionalized dibenzhydryl substituents ([Fig polymers-12-02509-ch001]F) were designed, synthesized, characterized, and investigated in ethylene polymerization and copolymerization with methyl acrylate. On account of distance, the *ortho*-substituents may lead to a more significant improvement of the steric hindrance effect as well as larger impact on the performance of the catalysts compared with the corresponding *meta*- and *para*-substituents [63,64,65,66,67,68,69,70,71]. Consequently, this may lead to catalyst with enhanced thermal stabilities. Moreover, the *ortho*-functionalization provides an approach to install functional moieties to the structures of the catalysts. For instance, the *ortho*-hydroxyl functionalized dibenzhydryl substituents are Brønsted acidic moieties. The hydroxyl groups may form Brønsted acid-base interactions with some polar monomers and improve the performances of the catalysts in ethylene copolymerization. Chen et al. reported that the ligand–substrate effect can improve the catalytic performances of the phosphine-sulfonate palladium and nickel catalysts bearing Brønsted basic moieties in the copolymerization of ethylene with Brønsted acidic polar monomers [3,72]. Moreover, Jordan et al. reported that the *α*-diimine Pd catalyst bearing the Brønsted acidic secondary amide moiety resulted in a higher incorporation of polar monomers when compared with the analogue bearing the tertiary amide unit in the ethylene copolymerizations with methyl acrylate as well as acrylic acid [73], indicating the ligand–substrate effect based on the Brønsted acid-base interaction.

## 2. Experimental

### 2.1. Materials

All of the air sensitive experiments were performed under nitrogen atmosphere while using a Schlenk line and glove box. Deuterated solvents that were used for NMR were dried and distilled prior to use. Dichloromethane was pre-dried overnight with Na_2_SO_4_ and then stirred with CaH_2_ under dry nitrogen for 10–12 h. Subsequently, it was distilled under inert conditions. Toluene was dried with sodium metal while using benzophenone as an indicator and distilled under nitrogen.

### 2.2. Methods

(COD)PdMeCl (COD = 1,5-cyclooctadiene) [74] and sodium tetrakis(3,5-bis[trifluoromethyl]phenyl)borate (NaBArF) [75] were prepared according to previously reported procedures, respectively. Other solvents and chemicals were purchased from Energy Chemicals and J&K Chemical (Beijing, China) and used as received. ^1^H, ^13^C, ^1^H ^1^H COSY, and ^1^H ^13^C COSY NMR spectra were recorded by a Bruker Ascend Tm 400 spectrometer (Bruker, Rheinstetten, Germany). Chemical shifts were referenced to a tetramethylsilane signal (0 ppm). Elemental analyses were performed on a VarioELIII instrument (Shanghai, China). Mass spectra were obtained while using electrospray ionization (ESI) LCMS-2010A (Shimadzu, Japan). Matrix-assisted laser desorption ionization-time of flight mass spectroscopy (MALDI-TOFMS) was performed on a Bruker ultrafleXtreme (Bremen, Germany). X-ray diffraction data were collected at 298(2) K on a Bruker Smart CCD area detector (Bremen, Germany) with graphite-monochromated MoKα (λ = 0.71073 nm). Molecular weight and molecular weight distribution of the polymer were determined by gel permeation chromatography (GPC) with a PL320 that was equipped with two Agilent PLgel Olexis columns at 150 °C using *o*-dichlorobenzene as the solvent. The calibration was generated using a polystyrene standard and it was corrected for linear polyethylene by universal calibration using the Mark–Houwink parameters of Rudin: K = 1.75 × 10^−2^ cm^3^/g and R = 0.67 for polystyrene and K = 5.90 × 10^−2^ cm^3^/g and R = 0.69 for polyethylene.

### 2.3. Preparation of ***1a*** and ***1b***

The compounds **1a** and **1b** were synthesized by following the literatures procedures [76].

### 2.4. General Procedure for the Synthesis of ***2*** and ***3***

A mixture of 2,4-dimethylaniline (1.21 g, 0.01 mol) and alcohols **1a** (2.44 g, 0.01 mol) while alcohol **1b** (2.14 g, 0.01 mol) and 12.8 g of LiClO_4_·3H_2_O in 20 mL of CH_3_NO_2_ (4M soln.) were heated separately at 80 °C for 10–12 h until the completion of reaction, as monitored by TLC. After the completion of reaction, the solvent was removed under vacuum and the crude product was purified by column chromatography with 1:8 EtOAc/ petroleum ether to afford aniline **2** as an off-white solid while aniline **3** as a light yellowish solid.

*Preparation of **2**.* Off-white solid, Yield: 78%, ^1^H NMR (CDCl_3_, 400 MHz): *δ* 7.25–7.20 (m, 2H, *Ar*), 6.8920136.83 (m, 6H, *Ar*), 6.77 (d, *J* = 4 Hz, 1H, *Ar*), 6.33 (d, *J* = 4 Hz, 1H, *Ar*), 6.12 (s, 1H, C*H*Ph_2_), 3.72 (s, 3H, Ar-OC*H*_3_), 3.07 (*br* s, 2H, N*H*_2_), 2.13 (s, 3H, Ar-C*H*_3_), 2.10 (s, 3H, Ar-C*H*_3_).^13^C NMR (101 MHz, CDCl_3_): *δ* 157.39, 139.51, 131.09, 130.07, 129.06, 128.61, 127.51, 127.37, 126.58, 122.31, 120.25, 110.78, 55.83 (Ar-O*C*H_3_), 37.91 (*C*HPh_2_), 20.75 (Ar-*C*H_3_), 17.79 (Ar-*C*H_3_). ESI-MS (*m*/*z*): 348.19 [M + H]^+^.

*Preparation of **3**.* Light yellowish solid, Yield: 75%, ^1^H NMR (CDCl_3_, 400 MHz): *δ* 7.29–7.19 (m, 4H, *Ar*), 7.12–7.10 (m, 2H, *Ar*), 6.92–6.86 (m, 3H, *Ar*), 6.80 (d, *J* = 4 Hz, 1H, *Ar*), 6.35 (d, *J* = 4 Hz, 1H, *Ar*), 5.85 (s, 1H, C*H*Ph_2_), 3.74 (s, 3H, Ar-OC*H*_3_), 3.15 (*br* s, 2H, N*H*_2_), 2.14 (s, 3H, Ar-C*H*_3_), 2.11 (s, 3H, Ar-C*H*_3_). ^13^C NMR (101 MHz, CDCl_3_): *δ* 157.14, 142.86, 139.63, 130.77, 130.32, 129.55, 129.29, 128.84, 128.24, 128.09, 127.84, 126.83, 126.23, 122.54, 120.50, 110.63, 55.71 (Ar-O*C*H_3_), 44.64 (*C*HPh_2_), 20.74 (Ar-*C*H_3_), 17.81 (Ar-*C*H_3_). ESI-MS (*m*/*z*): 318.18 [M + H]^+^.

### 2.5. General Procedure for the Synthesis of ***4*** and ***5***

The anilines **2** and **3** (1mol equiv.) were separately dissolved in appropriate amount of dry DCM under nitrogen and the mixtures were cooled to −40 °C. After this BBr_3_ (2.5 mol equiv.) was added dropwise and then the mixture was stirred overnight at room temperature. Finally, the reaction mixtures were quenched very cautiously with ice pieces and 5% NaHCO_3_ solution and extracted with DCM. The solvent was rotary evaporated to afford anilines **4** and **5** in good yield.

*Preparation of **4**.* Light yellow solid, Yield: 95%, ^1^H NMR (CDCl_3_, 400 MHz): *δ* 7.18–7.14 (m, 2H, *Ar*), 6.91–6.82 (m, 7H, *Ar*), 6.51 (d, *J* = 4 Hz, 1H, *Ar*), 5.83 (s, 1H, C*H*Ph_2_), 4.33 (*br* s, 4H, N*H*_2_ & O*H*), 2.13 (s, 3H, Ar-C*H*_3_), 2.09 (s, 3H, Ar-C*H*_3_). ^13^C NMR (101 MHz, CDCl_3_): *δ* 153.98, 138.92, 130.44, 129.95, 128.74, 128.45, 127.36, 127.30, 126.40, 124.04, 120.98, 116.56, 40.10 (*C*HPh_2_), 20.75 (Ar-*C*H_3_), 17.71 (Ar-*C*H_3_). ESI-MS (*m*/*z*): 320.16 [M + H]^+^.

*Preparation of **5**.* Light yellow solid, Yield: 96%, ^1^H NMR (CDCl_3_, 400 MHz): *δ* 7.31–7.23 (m, 4H, *Ar*), 7.14–7.09 (m, 3H, *Ar*), 6.84–6.81 (m, 3H, *Ar*), 6.47 (s, 1H, *Ar*), 5.67 (s, 1H, C*H*Ph_2_), 4.32 (*br* s, 2H, N*H*_2,_ 1H, -O*H*), 2.12 (s, 3H, Ar-C*H*_3_), 2.09 (s, 3H, Ar-C*H*_3_). ^13^C NMR (101 MHz, CDCl_3_): *δ* 154.03, 141.43, 138.56, 130.32, 130.09, 129.49, 128.91, 128.68, 128.64, 128.25, 128.17, 128.03, 126.82, 124.04, 120.72, 116.46, 46.41 (*C*HPh_2_), 20.76 (Ar-*C*H_3_), 17.76 (Ar-*C*H_3_). ESI-MS (*m*/*z*): 304.16 [M + H]^+^.

### 2.6. Procedure for the Synthesis of ***6***

A mixture of 2,6-diisopropylaniline (1mol equiv.) and 2,3-buten-dione (2.5 mol equiv.) was heated at 80 °C in toluene for 24 h until the completion of reaction was checked by TLC. After the completion of reaction the solvent was rotary evaporated and the product was recrystallized in *n*-Hexane in order to afford yellowish solid in excellent yield. Yellow solid, Yield: 86%, ^1^H NMR (CDCl_3_, 400 MHz): *δ* 7.17 (d, *J* = 8 Hz, 2H, *Ar*), 7.10 (t, *J* = 8 Hz, 1H, *Ar*), 2.71 (m, 2H, C*H*(CH_3_)_2_), 2.07 (s, 3H, C*H*_3_-C = O), 1.32–1.17 (m, 15H, CH(C*H*_3_)_2_ & C*H*_3_-C = N). ^13^C NMR (101 MHz, CDCl_3_): *δ* 168.20 (*C* = O), 146.18 (*C* = N), 135.08, 123.75, 123.01, 29.73 (*C*H_3_-C = O), 28.51 (*C*H(CH_3_)_2_), 23.01 (CH(*C*H_3_)_2_), 22.72 (CH(*C*H_3_)_2_), 16.60 (*C*H_3_-CN).

### 2.7. Procedures for the Syntheses of the Ligands

*Preparation of **L1***. Aniline **2** (1.2 mol equiv.) and compound **6** (1 mol equiv.) were heated in toluene using Dean-Stark apparatus in the presence of catalytic amount of *p*-toluenesulphonic acid. On completion of reaction, as checked by TLC, the solvent was rotatory evaporated and crude product was stirred in methanol overnight. The yellow precipitates were filtered and recrystallized in 2:1 DCM and methanol to afford pure product. Yellow solid, Yield: 76%, ^1^H NMR (CDCl_3_, 400 MHz): *δ* 7.20–7.11 (m, 4H, *Ar*), 7.05 (t, *J* = 8Hz, 1H, *Ar*), 6.90 (d, *J* = 4 Hz, 1H, *Ar*), 6.85–6.78 (m, 6H, *Ar*), 6.49 (d, *J* = 4 Hz, 1H, *Ar*), 6.05 (s, 1H, C*H*Ph_2_), 3.65 (s, 3H, Ar-OC*H*_3_), 3.61 (s, 3H, Ar-OC*H*_3_), 2.64–2.52 (m, 2H, C*H*(CH_3_)_2_), 2.20 (s, 3H, C*H*_3_-C = N), 1.95 (s, 3H, Ar-C*H*_3_), 1.88 (s, 3H, Ar-C*H*_3_), 1.66 (s, 3H, C*H*_3_-C = N), 1.21 (d, *J* = 4Hz, 3H, CH(C*H*_3_)_2_), 1.11 (dd, *J* = 4 Hz, 8 Hz, 9H, CH(C*H*_3_)_2_). ^13^C NMR (101 MHz, CDCl_3_): *δ* 168.21 (C = N), 168.16 (C = N), 157.52, 157.04, 146.38, 145.41, 135.03, 134.81, 132.00, 131.69, 130.79, 130.10, 128.88, 127.15, 127.03, 123.88, 123.47, 123.00, 122.69, 120.18, 111.08, 110.50, 56.00 (Ar-O*C*H_3_), 55.42 (Ar-O*C*H_3_), 38.84 (CHPh_2_), 28.39 (*C*H(CH_3_)_2_), 28.16 (*C*H(CH_3_)_2_), 23.04 (CH(*C*H_3_)_2_), 22.97 (CH(*C*H_3_)_2_), 22.68 (CH(*C*H_3_)_2_), 22.61 (CH(*C*H_3_)_2_), 21.08 (Ar-*C*H_3_), 17.57(Ar-*C*H_3_), 16.00 (C = N*C*H_3_), 15.95 (C = N*C*H_3_). ESI-MS (*m/z*): 575.36 [M + H]^+^.

*Preparation of **L2***. Aniline **3** (1.2 mol equiv.) and compound **6** (1 mol equiv.) were heated in toluene using Dean–Stark apparatus in the presence of catalytic amount of *p*-Toluenesulphonic acid. On completion of reaction, as checked by TLC, the solvent was rotatory evaporated and crude product was stirred in methanol overnight. The yellow precipitates were filtered and recrystallized in DCM to afford pure product. Yellow solid, Yield: 72%, ^1^H NMR (CDCl_3_, 400 MHz): *δ* 7.27–7.03 (m, 9H, *Ar*), 6.93–6.84 (m, 4H, *Ar*), 6.56–6.53 (m, 1H, *Ar*), 5.82, 5.69 (s, 1H, C*H*Ph_2_), 3.69, 3.62 (s, 3H, Ar-OC*H*_3_), 2.62–2.49 (m, 2H, C*H*(CH_3_)_2_), 2.22 (s, 3H, C*H*_3_-C = N), 1.99, 1.96 (s, 3H, Ar-C*H_3_*), 1.95, 1.91 (s, 3H, Ar-C*H*_3_), 1.57, 1.55 (s, 3H, C*H*_3_-C = N), 1.22 (dd, *J* = 4Hz, 8 Hz, 3H, CH(C*H*_3_)_2_), 1.14–1.08 (m, 9H, CH(C*H*_3_)_2_). ^13^C NMR (101 MHz, CDCl_3_): *δ* 168.67 (*C* = N), 167.99 (*C* = N), 157.13, 156.93, 146.27, 145.44, 143.42, 143.28, 135.01, 134.71, 132.75, 132.22, 132.02, 131.83, 130.80, 130.50, 129.87, 129.57, 129.13, 128.14, 127.96, 127.89, 127.40, 127.32, 126.87, 125.88, 124.15, 124.00, 123.60, 123.56, 123.01, 122.94, 122.74, 120.33, 120.13, 110.84, 110.41, 55.82 (Ar-O*C*H_3_), 55.37(Ar-O*C*H_3_), 45.87 (*C*HPh_2_), 45.32 (*C*HPh_2_), 28.46 (*C*H(CH_3_)_2_), 28.16 (*C*H(CH_3_)_2_), 28.03 (*C*H(C*H*_3_)_2_), 23.34 (CH(*C*H_3_)_2_), 23.06 (CH(*C*H_3_)_2_), 22.96 (CH(*C*H_3_)_2_), 22.93(CH(*C*H_3_)_2_), 22.71(CH(*C*H_3_)_2_), 22.59 (CH(*C*H_3_)_2_), 21.08 (Ar-*C*H_3_), 17.88 (Ar-*C*H_3_), 17.55 (Ar-*C*H_3_), 16.49 (C = N*C*H_3_), 16.07(C = N*C*H_3_), 15.96 (C = N*C*H_3_), 15.79 (C = N*C*H_3_). ESI-MS (*m*/*z*): 545.36 [M + H]^+^.

*Preparation of **L3***. Aniline **4** (1.2 mol equiv.) and compound **6** (1 mol equiv.) were heated in toluene and ethanol (1:1) keeping the temperature 105–110 °C in the presence of catalytic amount of *p*-toluenesulphonic acid. On completion of reaction as checked by TLC, the solvent was rotatory evaporated and crude product was stirred in small amount of DCM and excess of *n*-Hexane overnight. The yellow precipitates were filtered and recrystallized in ethanol in order to afford pure product. Yellow solid, Yield: 70%, ^1^H NMR (CDCl_3_, 400 MHz): *δ* 7.15–7.04 (m, 4H, *Ar*), 6.97–6.93 (m, 2H, *Ar*), 6.88–6.72 (m, 6H, *Ar*), 6.13 (*br*, s, 1H, -O*H*), 5.72 (s, 1H, C*H*Ph_2_), 5.42 (*br*, s, 1H, -O*H*), 2.60–2.45 (m, 2H, C*H*(CH_3_)_2_), 2.22 (s, 3H, C*H*_3_-C = N), 1.98 (s, 3H, Ar-C*H*_3_), 1.87 (s, 3H, Ar-C*H*_3_), 1.54 (s, 3H, C*H*_3_-C = N), 1.19 (d, *J* = 4 Hz, 3H, CH(C*H*_3_)_2_) 1.09 (dd, *J* = 4Hz, 8Hz, 9H, CH(C*H*_3_)_2_). ^13^C NMR (101 MHz, CDCl_3_): *δ* 170.36 (C = N), 167.71(C = N), 153.87, 153.83, 145.93, 144.77, 134.99, 134.91, 133.39, 130.93, 130.38, 129.92, 129.85, 129.42, 128.24, 128.05, 128.00, 127.71, 127.49, 125.36, 123.80, 123.00, 122.92, 120.70, 116.28, 116.25, 40.67 (*C*HPh_2_), 28.54 (*C*H(CH_3_)_2_), 27.96 (*C*H(CH_3_)_2_), 23.39 (CH(*C*H_3_)_2_), 22.91 (CH(*C*H_3_)_2_), 22.84 (CH(*C*H_3_)_2_), 22.74 (CH(*C*H_3_)_2_), 21.04 (Ar-*C*H_3_), 17.62 (Ar-*C*H_3_), 16.32 (C = N*C*H_3_), 16.28 (C = N*C*H_3_). ESI-MS (*m*/*z*): 547.33 [M + H]^+^.

*Preparation of **L4***. Aniline **5** (1.2 mol equiv.) and compound **6** (1 mol equiv.) were refluxed in toluene and ethanol (1:1) keeping the temperature 105–110 °C in the presence of catalytic amount of *p*-toluenesulphonic acid. On completion of reaction, as checked by TLC, the solvent was rotatory evaporated and the crude product was stirred in small amount of DCM and excess of *n*-hexane overnight. The yellow precipitates were filtered and recrystallized in DCM to afford pure product. Yellow solid, Yield: 68%, ^1^H NMR (CDCl_3_, 400 MHz): *δ* 7.23–7.13 (m, 3H, *Ar*), 7.09–6.97 (m, 6H, *Ar*), 6.92–6.68 (m, 5H, *Ar*), 5.57, 5.41 (s, 1H, C*H*Ph_2_), 2.55–2.39 (m, 2H, C*H*(CH_3_)_2_), 2.18, 2.17 (s, 3H, C*H*_3_-C = N), 1.98, 1.92 (s, 3H, Ar-C*H*_3_), 1.89, 1.87 (s, 3H, Ar-C*H*_3_), 1.46, 1.40 (s, 3H, C*H*_3_-C = N), 1.14 (dd, *J* = 4 Hz, 8 Hz, 3H, CH(C*H*_3_)_2_), 1.08–1.02 (m, 9H, CH(C*H*_3_)_2_). ^13^C NMR (101 MHz, CDCl_3_): *δ* 170.33 (*C* = N), 167.88 (*C* = N), 167.51(*C* = N), 154.05, 153.98, 145.99, 145.11, 142.03, 141.66, 141.38, 139.59, 135.01, 134.93, 133.23, 132.81, 132.03, 131.18, 130.39, 130.30, 130.11, 129.79, 129.68, 129.44, 129.32, 128.93, 128.83, 128.72, 128.43, 128.38, 128.18, 127.92, 127.88, 127.85, 127.20, 126.86, 126.46, 125.07, 124.89, 123.83, 123.72, 123.45, 123.03, 122.98, 120.79, 120.68, 120.36, 116.43, 116.24, 46.86 (CHPh_2_), 46.17 (CHPh_2_), 28.53 (*C*H(CH_3_)_2_), 28.12 (*C*H(CH_3_)_2_), 27.92 (*C*H(CH_3_)_2_), 23.44 (CH(*C*H_3_)_2_), 23.34 (CH(*C*H_3_)_2_), 22.98 (CH(*C*H_3_)_2_), 22.90 (CH(*C*H_3_)_2_), 22.82 (CH(*C*H_3_)_2_), 22.75 (CH(*C*H_3_)_2_), 22.70 (CH(*C*H_3_)_2_), 21.06 (CH(*C*H_3_)_2_), 20.69 (Ar-*C*H_3_), 17.76(Ar-*C*H_3_), 17.62 (Ar-*C*H_3_), 17.58 (Ar-*C*H_3_), 16.66(C = NCH_3_), 16.37(C = NCH_3_), 16.29(C = NCH_3_), 15.95 (C = NCH_3_). ESI-MS (*m*/*z*): 531.33 [M + H]^+^.

### 2.8. General Procedure for the Synthesis of the Pd Catalysts

The ligands **L1**, **L2**, **L3**, **L4** (1mol equiv.) and (COD)PdMeCl were stirred in a suitable amount of dry DCM in glove box for 3-4 days at room temperature. During stirring, the ligands were completely dissolved and the color of the solution was changed from yellow to red. At the end of the reaction, the desired compound was isolated while using column chromatography. The mixture was eluted on silica gel with first 1:1 hexane/CH_2_Cl_2_, and then pure CH_2_Cl_2_ as the mobile phase.

*Preparation of **Pd1***. Yellow solid, Yield: 88%, ^1^H NMR (CDCl_3_, 400 MHz): *δ* 7.26–7.18 (m, 5H, *Ar*), 7.06–6.96 (m, 3H, *Ar*), 6.91–6.82 (m, 5H, *Ar*), 6.19, 6.01 (s, 1H, C*H*Ph_2_), 3.79, 3.75 (s, 3H, Ar-OC*H*_3_), 3.58, 3.54 (s, 3H, Ar-OC*H*_3_), 3.18–3.08, 3.05–2.94 (m, 2H, C*H*(CH_3_)_2_), 2.32, 2.27 (s, 3H, Ar-C*H*_3_), 2.23, 2.18 (s, 3H, Ar-C*H*_3_), 1.78, 1.72 (s, 3H, C*H*_3_-C = N), 1.57–1.20 (m, 9H, CH(C*H*_3_)_2_), 1.10–1.07 (m, 6H, CH(C*H*_3_)_2_ & C*H*_3_-C = N), 0.60, 0.45 (s, 3H, C*H*_3_Pd). ^13^C NMR (101 MHz, CDCl_3_): *δ* 174.82 (*C* = N), 174.01(*C* = N), 169.73 (*C* = N), 169.11 (*C* = N), 158.11, 157.75, 157.51, 157.34, 141.68, 141.63, 141.40, 141.24, 138.89, 138.51, 138.08, 137.81, 135.23, 134.68, 133.23, 132.51, 131.96, 131.78, 131.52, 131.07, 130.63, 130.37, 130.00, 129.18, 128.68, 128.09, 128.05, 127.93, 127.73, 127.57, 127.48, 127.40, 126.84, 123.93, 123.84, 123.25, 123.18, 121.10, 121.07, 120.35, 119.56, 113.51, 112.93, 111.00, 110.61, 56.96 (Ar-O*C*H_3_), 55.98 (Ar-O*C*H_3_), 55.18 (Ar-O*C*H_3_), 40.28 (*C*HPh_2_), 28.90 (*C*H(CH_3_)_2_), 28.56 (*C*H(CH_3_)_2_), 28.30 (*C*H(CH_3_)_2_), 27.91 (*C*H(CH_3_)_2_), 24.22 (CH(*C*H_3_)_2_), 24.16 (CH(*C*H_3_)_2_), 23.82 (CH(*C*H_3_)_2_), 23.68 (CH(*C*H_3_)_2_), 23.51(CH(*C*H_3_)_2_), 23.48 (CH(*C*H_3_)_2_), 23.19(CH(*C*H_3_)_2_), 21.33 (Ar-*C*H_3_), 21.19 (Ar-*C*H_3_), 20.60 (Ar-*C*H_3_), 19.66 (Ar-*C*H_3_), 19.02 (*C*H_3_-CN), 18.99 (*C*H_3_-CN), 18.89 (*C*H_3_-CN), 17.92 (*C*H_3_-CN), 2.65 (*C*H_3_Pd), 2.16 (*C*H_3_Pd). Key C-H HSQC: *δ* 6.19/40.28 (*CH*Ph_2_), 3.79/56.97 (Ar-OC*H_3_*), 3.58/55.18 (Ar-OC*H_3_*), 3.12/28.71 (*CH*(CH_3_)_2_), 2.27/21.33 (Ar-*CH*_3_), 2.18/20.60 (Ar-C*H_3_*), 1.72/18.99 (*CH*_3_-CN), 1.10/17.99 (*CH*_3_-CN), 0.60/2.16 (*CH*_3_Pd). MALDI-TOF-MS (*m*/*z*): 766.57 [M + HCl]^+^. Anal. Calcd. for C_40_H_49_ClN_2_O_2_Pd: C, 65.66, H, 6.75, N, 3.83 Found: C, 65.62, H, 6.71, N, 3.79.

*Preparation of **Pd2***. Yellowish solid, Yield: 89%, ^1^H NMR (CDCl_3_, 400 MHz): *δ* 7.31–7.09 (m, 10H, *Ar*), 7.00–6.84 (m, 4H, *Ar*), 6.21, 6.16, 6.02 (s, 1H, C*H*Ph_2_), 3.91, 3.83, 3.79, 3.60, 3.59, 3.53 (s, 3H, Ar-OC*H*_3_), 3.28–3.21, 3.16–3.02, 2.98–2.90 (m, 2H, C*H*(CH_3_)_2_), 2.28, 2.26, (s, 3H, Ar-C*H*_3_), 2.23, 2.14 (s, 3H, Ar-C*H*_3_), 1.74, 1.73, 1.66 (s, 3H, C*H*_3_-C = N), 1.56 1.04 (m, 12H, CH(C*H*_3_)_2_), 0.97, 0.90, 0.88 (s, 3H, C*H*_3_-C = N), 0.65, 0.58, 0.54 (s, 3H, C*H*_3_Pd). ^13^C NMR (101 MHz, CDCl_3_): *δ* 175.55 (*C* = N), 175.28 (*C* = N), 174.35 (*C* = N), 174.27 (*C* = N), 170.63 (*C* = N), 169.49 (*C* = N), 169.19 (*C* = N), 158.46, 158.32, 157.09, 143.23, 142.55, 141.68, 141.54, 141.50, 141.47, 141.19, 138.76, 138.72, 138.56, 138.07, 137.98, 137.95, 137.77, 135.86, 135.57, 134.97, 134.76, 134.62, 134.44, 133.78, 132.79, 132.10, 131.98, 131.78, 130.96, 130.82, 130.75, 130.53, 130.34, 130.30, 129.83, 129.74, 129.48, 129.41, 129.39, 129.32, 129.25, 129.06, 128.64, 128.47, 128.45, 128.43, 128.11, 128.03, 127.94, 127.89, 127.74, 127.66, 127.63, 127.60, 127.20, 127.00, 126.92, 126.88, 126.62, 126.37, 126.32, 125.90, 123.99, 123.95, 123.86, 123.31, 123.30, 123.21, 121.16, 121.07, 119.57, 119.31, 114.04, 113.46, 110.87, 110.70, 100.90, 57.49 (Ar-O*C*H_3_), 56.90 (Ar-O*C*H_3_), 55.49 (Ar-O*C*H_3_), 55.04 (Ar-O*C*H_3_), 47.22 (*C*HPh_2_), 46.50 (*C*HPh_2_), 45.43 (*C*HPh_2_), 31.05 (*C*HPh_2_), 28.96 (*C*H(CH_3_)_2_), 28.94 (*C*H(CH_3_)_2_), 28.87 (*C*H(CH_3_)_2_), 28.62 (*C*H(CH_3_)_2_), 28.42 (*C*H(CH_3_)_2_), 28.33 (*C*H(CH_3_)_2_), 28.01 (*C*H(CH_3_)_2_), 27.66 (*C*H(CH_3_)_2_), 24.37 (CH(*C*H_3_)_2_), 24.26 (CH(*C*H_3_)_2_), 24.06 (CH(*C*H_3_)_2_), 23.96 (CH(*C*H_3_)_2_), 23.89 (CH(*C*H_3_)_2_), 23.85 (CH(*C*H_3_)_2_), 23.80(CH(*C*H_3_)_2_), 23.58(CH(*C*H_3_)_2_), 23.50 (CH(*C*H_3_)_2_), 23.48(CH(*C*H_3_)_2_), 23.35 (CH(*C*H_3_)_2_), 23.16(CH(*C*H_3_)_2_), 23.01(CH(*C*H_3_)_2_), 21.36 (Ar-*C*H_3_), 21.27(Ar-*C*H_3_), 21.24 (Ar-*C*H_3_), 21.15 (Ar-*C*H_3_), 20.62 (Ar-*C*H_3_), 20.47 (Ar-*C*H_3_), 19.50 (Ar-*C*H_3_), 19.29 (Ar-*C*H_3_), 19.13 (*C*H_3_-C = N), 19.00 (*C*H_3_-C = N), 18.62 (*C*H_3_-C = N), 18.32 (*C*H_3_-C = N), 18.21 (*C*H_3_-C = N), 17.78 (*C*H_3_-C = N), 17.64 (*C*H_3_-C = N), 17.56 (*C*H_3_-C = N), 3.00 (*C*H_3_Pd), 2.94(*C*H_3_Pd), 2.53 (*C*H_3_Pd), 2.19 (*C*H_3_Pd). MALDI-TOF-MS (*m*/*z*): 738.61 [M + 3H,Cl] ^+^. Anal. Calcd. for C_39_H_47_ClN_2_OPd: C, 66.76, H, 6.75, N, 3.99 Found: C, 66.70, H, 6.69, N, 3.93.

*Preparation of **Pd3***. Redish solid, Yield: 78%, ^1^H NMR (CDCl_3_, 400 MHz): *δ* 7.32–7.13 (m, 5H, *Ar*), 7.05–7.00 (m, 3H, *Ar*), 6.85–6.82 (m, 5H, *Ar*), 6.12 (*br*, s, 1H, -O*H*), 5.57 (s, 1H, C*H*Ph_2_), 3.15–2.98 (m, 2H, C*H*(CH_3_)_2_), 2.27 (s, 3H, Ar-C*H*_3_), 2.09 (s, 3H, Ar-C*H*_3_), 1.71 (s, 3H, C*H*_3_-C = N), 1.39 (d, *J* = 8 Hz, 3H, CH(C*H*_3_)_2_), 1.30–1.24 (m, 9H, CH(C*H*_3_)_2_ & C*H*_3_-C = N). 1.05 (d, *J* = 8 Hz, 3H, CH(C*H*_3_)_2_), 0.57 (s, 3H, C*H*_3_Pd). ^13^C NMR (101 MHz, CDCl_3_): *δ* 174.98 (*C* = N), 171.35(*C* = N), 155.09, 141.29, 138.95, 138.56, 131.55, 130.72, 130.07, 128.80, 128.37, 127.89, 124.22, 123.86, 120.51, 120.21, 118.50, 29.71 (*C*HPh_2_), 28.57 (*C*H(CH_3_)_2_), 28.09 (*C*H(CH_3_)_2_), 24.23 (CH(*C*H_3_)_2_), 23.83(CH(*C*H_3_)_2_), 23.81(CH(*C*H_3_)_2_), 22.89 (CH(*C*H_3_)_2_), 21.30 (Ar-*C*H_3_), 20.52 (Ar-*C*H_3_), 19.06 (*C*H_3_-C = N), 17.59 (*C*H_3_-C = N), 4.38 (*C*H_3_Pd) MALDI-TOF-MS (*m*/*z*): 738.78 [M +HCl]^+^. Anal. Calcd. for C_38_H_45_ClN_2_O_2_Pd: C, 64.86, H, 6.45, N, 3.98 Found: C, 65.72, H, 6.38, N, 3.90.

*Preparation of **Pd4***. Redish solid, Yield: 68%, ^1^H NMR (CDCl_3_, 400 MHz): *δ* 7.32–7.16 (m, 9H, *Ar*), 7.03–6.97 (m, 3H, *Ar*), 6.78–6.73 (m, 2H, *Ar*), 5.99 (s, 1H, C*H*Ph_2_), 3.17–2.96 (m, 2H, C*H*(CH_3_)_2_), 2.26 (s, 3H, Ar-C*H*_3_), 2.08 (s, 3H, Ar-C*H*_3_), 1.68 (s, 3H, C*H*_3_-C = N), 1.46 (d, *J* = 4 Hz, 3H, CH(C*H*_3_)_2_), 1.33–1.26 (m, 6H, CH(C*H*_3_)_2_), 1.06 (d, *J* = 4 Hz, 3H, CH(C*H*_3_)_2_), 1.02 (s, 3H, C*H*_3_-C = N), 0.57 (s, 3H, C*H*_3_Pd). ^13^C NMR (101 MHz, CDCl_3_): *δ* 174.80 (*C* = N), 171.13 (*C* = N), 155.27, 142.87, 141.45, 141.32, 138.61, 138.45, 135.99, 132.18, 130.69, 130.20, 130.17, 128.54, 128.30, 127.87, 127.27, 126.84, 126.77, 126.60, 124.00, 119.22, 117.80, 48.03 (*C*HPh_2_), 28.51 (*C*H(CH_3_)_2_), 28.49 (*C*H(CH_3_)_2_), 24.17 (CH(*C*H_3_)_2_), 23.79 (CH(*C*H_3_)_2_), 23.59 (CH(*C*H_3_)_2_), 22.97 (CH(*C*H_3_)_2_), 21.28 (Ar-*C*H_3_), 20.46 (Ar-*C*H_3_), 18.58 (*C*H_3_-C = N), 17.52 (*C*H_3_-C = N), 4.20 (*C*H_3_Pd) Key C-H HSQC: *δ* 5.99/48.03 (*CH*Ph_2_), 3.17–2.96/28.51–28.49 (m, 2H, *CH*(CH_3_)_2_), 2.26/21.28 (Ar-*CH*_3_), 2.08/20.46 (Ar-*CH*_3_), 1.68/18.58 (*CH*_3_-C = N), 1.02/17.52 (*CH*_3_-C = N), 0.57/4.20 (*CH*_3_Pd). MALDI-TOF-MS (*m*/*z*): 738.61 [M-CH_3_,Cl]^+^. Anal. Calcd. for C_38_H_45_ClN_2_OPd: C, 66.37, H, 6.60, N, 4.07 Found: C, 66.29, H, 6.59, N, 4.03.

### 2.9. General Procedure for Ethylene Polymerization

In a Glovebox, a 10 mL glass thick-walled pressure vessel was charged with sodium tetrakis(3,5-bis(trifluoromethyl)phenyl)borate (NaBAF, 1.2 mol eq.) in 2 mL toluene and a magnetic bar. The vessel was pressurized with ethylene and then allowed to equilibrate under constant pressure for a few minutes with stirring. Pre-catalysts (1 mol eq.) (**Pd1**), (**Pd3**), **(Pd2)**, and **(Pd4)** were injected to initiate polymerization respectively and continuously stirred for the desired time under specific temperature. The polymerization was quenched via the addition of MeOH and the polymer was precipitated while using excess MeOH and dried in a vacuum oven to a constant weight. Branch density (BD) was analyzed by using ^1^HNMR spectrum: BD = 1000 × (2/3) × (*I*_CH3_)/(*I*_CH2 and CH_ + *I*_CH3_). *δ* (ppm): CH_3_ (m, 0.77–0.95), CH_2_ and CH (m, ca. 1.0–1.45) [46].

### 2.10. General Procedure for Pd Catalyzed Ethylene Copolymerization

A 10 mL glass thick-walled pressure vessel was heated at 150 °C for 2–3 h under vacuum and then cooled to room temperature. The flask was pressurized with 3-atm ethylene and evacuated three times. 2 M solution of polar monomer (MA) and NaBAF (1.2 mol eq.) were introduced into the flask with an appropriate amount of toluene followed by the addition of **Pd1**–**4** catalyst (30 μmol) in DCM under ethylene atmosphere. Ethylene pressure was maintained at 3-atm for 5 h by continuously feeding ethylene. Finally, the polymerization was terminated by the addition of methanol. Subsequently, the oily polymers were further washed with methanol and then dried in vacue at 50 °C. The incorporation of MA was calculated by ^1^H NMR spectrum. The branches ending with the functional group were added to the total branches. MA% = 4 *I*_OMe_/3(*I*_CH3_ + *I*_CH2_ + *I*_CH_) × 100%. OMe (s, ca. 3.61–3.76 ppm); CH_2_ and CH (m, ca. 1.0–1.45 ppm); CH_3_ (m, 0.77–0.95 ppm) [46].

## 3. Results and Discussion

The anilines **2** and **3** were synthesized by the reactions of 2,4-dimethylaniline with 1,1-bis(2-methoxyphenyl)methanol or 1-(2-methoxyphenyl)-1-phenylmethanol in a solution of trihydrated lithiumperchlorate in nitromethane at 80 °C in 70–95% yields (Scheme 1) [76]. Demethylation was carried out by treating with boron tribromide at −40 °C in dichloromethane, forming the anilines **4** and **5** in 95% and 96% yields, respectively. The demethylation of anilines was confirmed by ^1^H NMR with the disappearance of methoxy signals and appearance of signals of hydroxyl proton at 4.3 ppm in CDCl_3_. Compound **6** was synthesized by heating 2,6-diisopropylaniline with 2,3-butandione in toluene at 80 °C. Ligands **L1** and **L2** were synthesized by refluxing **6** with the anilines **2** and **3** in toluene that was catalyzed by *p*-toluenesulfonic acid (PTSA). The ligand **L3** and **L4** were synthesized by heating the mixture of **6** with **4** or **5** in toluene and ethanol (1:1) at 105 °C in the presence of PTSA (Scheme 1). All of the ligands were characterized by ^1^H NMR, ^13^C NMR and ESI-MS. **L1** and **L3** were detected as a single isomer with single set of NMR peaks. **L2** and **L4** exhibit *cis* and *trans* isomers with two sets of peaks having different intensities. The more intense peaks may belong to *trans*-isomer due to its greater stability than *cis*-isomer at room temperature.

The palladium complexes were prepared by stirring the ligands with (COD)PdMeCl (COD = 1,5-cyclooctadiene) in dichloromethane (DCM) for 3–4 days at room temperature (Scheme 1) [46,47,48,49,50]. The formations of the metal complexes were confirmed by ^1^H NMR, ^13^C NMR, COSY, HSQC, and MALDI-TOF-MS. **Pd1** was observed in two isomeric forms with respect to the orientation of methyl and chloride attached with palladium in unsymmetrical *α*-diimine system. Both of these isomers were detected in ^1^H NMR, ^13^C NMR, COSY, and HSQC spectra. For simplicity in ^1^H NMR spectrum assignment, two sets of peaks were treated for a single isomer, because many peaks were in overlapped form and also with different intensities indicating different proportion of isomers, but the differentiated peaks have been assigned with their corresponding values. Furthermore, the MALDI-TOF confirmed these are the isomers of same catalyst **Pd1** with single mass value (Appendix A). Catalyst **Pd2** has chiral center in mono-substituted dibenzhydryl unit, with racemic (*RS*) configuration. The orientation of methyl and chloride attached with palladium makes more isomers. Therefore, the ^1^H NMR and ^13^C NMR spectra of this catalyst were massive (Appendix A). Although we separated some of these isomers using preparative TLC and compared the ^1^H NMR and ^13^C NMR spectra of these isomers, we were not able to separate them in complete proportions. The MALDI-TOF analysis confirmed that these are the isomers of same catalyst **Pd2** with single mass value. On the other hand, catalysts **Pd3** and **Pd4** with hydroxyl substitution pattern were detected as single isomers in ^1^H NMR and ^13^CNMR spectra (Appendix A).

Single crystals of catalysts **Pd1** and **Pd2** were obtained in DCM with the layering of *n*-hexane while complex **Pd3** and **Pd4** also in DCM, but with the slow diffusion of diethyl ether. The molecular structures were determined by X-ray diffraction (Figure 1), with observed bond lengths and bond angles typically for previously reported α-diimine palladium complexes [42,43,44,45,46,47,48,49,50]. The palladium center with square planner geometry has been observed in all of the complexes. The distance from metal center to oxygen of methoxy group was recorded 4.27 and 4.43 Å in **Pd1** and **Pd2**, respectively. In **Pd3** and **Pd4**, the distance between hydrogen of hydroxyl and chloride attached to palladium was found 2.49 and 2.33 Å, respectively, confirming the perfect intramolecular H-bonding. Furthermore, the N-Pd-N bite angle was comparable in all of catalysts, ranging from 77.4° to 77.9°.

By the activation of 1.2 equiv. NaBAF, the α-diimine Pd catalysts **Pd1**–**Pd4** were found to be highly active in ethylene polymerization (Table 1, entries 1–12). Both catalytic activity and molecular weight increased significantly, along with the increasing number of *ortho*-substituents (Table 1, entries 1–3 versus 4–6, entries 7–9 versus 10–12), indicating that the *ortho*-substituents can lead to the significant improvement of steric-hindrance and improve the catalytic performance of the catalysts [42,43,44,45]. All of these catalysts exhibit high activities of up to 8.8 × 10^5^ g PE (mol Pd h)^−1^ at 80 °C, and high polymer *M*_n_ (up to 3.93 × 10^4^ g mol^−1^) in ethylene polymerization. The activities and molecular weights decreased less than an order of magnitude at 80 °C compared with at 40 °C. These results indicated that the sterically bulky *ortho*-functionalized dibenzhydryl groups can prevent the N-aryl bonds from rotation and, consequently, improve the thermal stabilities of the catalysts. The activities and molecular weights were comparable with the previously reported *α*-diimine Pd catalysts bearing four dibenzhydryl groups. However, the catalysts **Pd1**~**Pd4** produced amorphous oily polyethylenes with high branching densities, while the *α*-diimine Pd catalysts bearing four dibenzhydryl groups produced semi-crystalline products with low branching densities [46,47,48,49,50]. The previously reported isopropyl-based α-diimine Pd catalysts bearing a dinaphthobarrelene-backbone or a nitrogen-containing second-coordination-sphere exhibited relatively low catalytic activities at 80 °C, and generated highly branched polyethylenes with low molecular weight and broad molecular weight distributions [55,77]. The catalysts (**Pd3** and **Pd4**) bearing hydroxyl groups showed similar activities as well as molecular weights of polyethylenes at 40 °C when compared with the catalysts (**Pd1** and **Pd2**) bearing methoxyl groups. They resulted in higher branching densities than those of their simulants bearing methoxyl groups, indicating that the methoxyl groups exhibit increased steric-hinerance effect when compared with the hydroxyl groups.

All of these Pd catalysts exhibit high activities in ethylene copolymerization with methyl acrylate (MA) (Table 2, entries 1–4). The incorporation ratios of MA decreased with the number of *ortho*-substituents increasing (Table 2, entries 1 versus 2–4), which indicated that the steric-hindrance effect that was associated with the *ortho*-substituents prevents the polar monomer from copolymerization reaction. The catalysts (**Pd3** and **Pd4**) bearing hydroxyl groups resulted in slightly higher incorporation ratios of MA compared with the catalysts (**Pd1** and **Pd2**) bearing methoxyl groups (Table 2, entries 1 versus 3, and 2 versus 4). This could be due to the ligand-substrate effect based on the Brønsted acid-base interaction [73]. Nevertheless, it can also be due to the less steric-hinerance effect that is associated with the hydroxyl groups [57].

## 4. Conclusions

In summary, a series of α-diimine Pd catalysts bearing *ortho*-methoxyl/hydroxyl functionalized dibenzhydryl substituents have been designed, synthesized, characterized, and investigated in ethylene polymerization and copolymerization with methyl acrylate. The catalytic activity and molecular weight of polyethylene product increased significantly with the increasing number of *ortho*-substituents, which suggested that the *ortho*-substituents resulted in the significant improvement of steric-hindrance associated with the improved catalytic performance. These Pd catalysts displayed remarkable thermal stability in ethylene polymerization, maintaining high activity of up to 8.8 × 10^5^ g PE (mol Pd h)^−1^ at 80 °C, and produced high molecular weight polyethylenes (*M*_n_ up to 3.93 × 10^4^ g mol^−1^), suggesting that the sterically bulky *ortho*-functionalized dibenzhydryl groups can keep the N-aryl bonds from rotation at high temperatures and improve the thermal stabilities of the catalysts. The Pd catalysts exhibit high activities in ethylene copolymerization with MA. The catalysts bearing hydroxyl groups resulted in slightly higher incorporation ratios of MA unit when compared with the catalysts bearing methoxyl groups. It is envisaged that *ortho*-functionalized dibenzhydryl substituents will be applicable to other catalytic systems for olefin polymerizations.

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
