# Peer review of "Ortho-Functionalized Dibenzhydryl Substituents in α-Diimine Pd Catalyzed Ethylene Polymerization and Copolymerization"

_polymers, 2020, doi:10.3390/polym12112509_

Round 1

Reviewer 1 Report

The article 'Ortho-functionalized dibenzhydryl substituents in α-diimine Pd catalyzed ethylene polymerization and copolymerization' raises still actual problem of the development of single-site catalysts of polymerization and copolymerization of α-olefins. The problem of the synthesis of copolymers based on monomers of different nature (α-olefin – polar vinyl substrate) can be solved by different approaches, the use of the late transition metal-based complexes represent a promising way, notwithstanding the relative low activity of these complexes in comparison with traditional Group 4 metal derivatives. In addition, the reaction mechanism of polymerization, catalyzed by Group 8 metal complexes, differs from the mechanisms of polymerization in the presence of traditional Ziegler-Natta catalysts, thereby enabling the synthesis of the materials with forward-looking characteristics. The manuscript under review contributes significantly to the further development of this topic, this solid experimentsl work is of interest for a wide range of scientists employed in the field of coordination polymerization of olefins, and therefore can be published in Polymers journal.

Comment:

The description of the Supplementary Materials is absent in the manuscript. Please, add this description in accordance with Polymers Instructions for Authors

Minor comments:

line 36 – 'preventing' is not good term

line 62, 390 – 'steric-hinerance'?

line 76 – Scheme 1 should be moved to page 4, the references to this scheme should be added in Section 2.3

line 105 – LiClO4·3H2O

line 287 – 10 mL

line 287 – please give decryption of NaBAF4 abbreviation here

line 312 – yields, respectively

line 314 – 2,6-diisopropylaniline

line 315 – 80 °C

line 361 – please remove decryption of NaBAF4 abbreviation

lines 367–369, 377, 400 – replace o by °

Author Response

Reviewer: 1

The article 'Ortho-functionalized dibenzhydryl substituents in α-diimine Pd catalyzed ethylene polymerization and copolymerization' raises still actual problem of the development of single-site catalysts of polymerization and copolymerization of α-olefins. The problem of the synthesis of copolymers based on monomers of different nature (α-olefin – polar vinyl substrate) can be solved by different approaches, the use of the late transition metal-based complexes represent a promising way, notwithstanding the relative low activity of these complexes in comparison with traditional Group 4 metal derivatives. In addition, the reaction mechanism of polymerization, catalyzed by Group 8 metal complexes, differs from the mechanisms of polymerization in the presence of traditional Ziegler-Natta catalysts, thereby enabling the synthesis of the materials with forward-looking characteristics. The manuscript under review contributes significantly to the further development of this topic, this solid experimentsl work is of interest for a wide range of scientists employed in the field of coordination polymerization of olefins, and therefore can be published in Polymers journal.

Comment:

The description of the Supplementary Materials is absent in the manuscript. Please, add this description in accordance with Polymers Instructions for Authors

Thanks for your useful suggestion. The description of the Supplementary Materials has been added according to your suggestions:

Supplementary Materials: Experimental details including synthesis and characterization of the complexes and (co)polymers, (co)polymerization data, and X-ray crystallographic data (PDF). CCDC. 1941586, 1957450, 1941588 and 1959442 contain the supplementary crystallographic data for this paper. Copies of the data can be obtained free of charge on application to CCDC, 12 Union Road, Cambridge CB2 1EZ, UK [Fax: (internat.) +44-1223/336-033; E-mail: deposit@ccdc.cam.ac.uk].

On the other hand catalysts Pd3 and Pd4 with hydroxyl substitution pattern were detected as single isomers in 1H NMR and 13CNMR spectra (Figures S28-32 from Supplementary Materials).

Minor comments:

line 36 – 'preventing' is not good term

Thanks for your useful suggestion.

"preventing their practical applications" has been corrected to " which has limited their practical applications".

line 62, 390 – 'steric-hinerance'?

Thanks a lot for catching this.

"steric-hinerance" has been corrected to "steric hindrance".

line 76 – Scheme 1 should be moved to page 4, the references to this scheme should be added in Section 2.3

Thanks for your useful suggestion. Scheme 1 has been moved in the results and discussion section.           (Scheme 1) [76]

line 105 – LiClO4·3H2O

"LiClO4.3H2O" has been corrected to "LiClO4·3H2O".

line 287 – 10 Ml

"10mL" has been corrected to "10 mL".

line 287 – please give decryption of NaBAF4 abbreviation here

sodium tetrakis(3,5-bis(trifluoromethyl)phenyl)borate (NaBAF, 1.2 mol eq.)

line 312 – yields, respectively

"yields respectively" has been corrected to "yields, respectively".

line 314 – 2,6-diisopropylaniline

"2,6-diisopropyl aniline" has been corrected to "2,6-diisopropylaniline".

line 315 – 80 °C

"80°C" has been corrected to "80 °C".

line 361 – please remove decryption of NaBAF4 abbreviation

NaBAF abbreviation has been removed.

lines 367–369, 377, 400 – replace by

lines 367–369: All of these catalysts exhibit high activities of up to 8.8 × 105  g PE (mol Pd h)−1 at 80 oC, and high polymer Mn (up to 3.93 × 104 g mol-1) in ethylene polymerization.

lines 377: …exhibited relatively low catalytic activities at 80 oC,and generated highly branched polyethylenes with low molecular weight and broad molecular weight distributions [55,77].

lines 400: These Pd catalysts displayed remarkable thermal stability in ethylene polymerization, maintaining high activity of up to 8.8 × 105  g PE (mol Pd h)−1 at 80 oC, and produced high molecular weight polyethylenes (Mn up to 3.93 × 104 g mol-1),…

Reviewer 2 Report

The manuscript by Chen and coworker reports Ortho-functionalized dibenzhydryl substituents in α-diimine Pd catalyzed ethylene polymerization and copolymerization have been found effective and worthy to report. There are some comments to be considered. 1: This type of work seems to be routine work by using such type of complexes. But, it contains an efficient catalytic system having good impact in polymerization reaction. 2. In scheme 1 the ligands L3 and L4 having substituents are R3 and R4, which are changes in palladium complexes R1 and R2 its very confusing. 3. For the thermal stability of all the complexes TGA data should be mention because palladium catalyst stability Generally, α-diimine palladium catalysts with ether auxiliary ligands for ethylene polymerization are prone to be deactivated at high temperature. 4. The morphology of the polymer should be characterized using SEM and TEM analysis.

Author Response

Reviewer: 2

The manuscript by Chen and coworker reports Ortho-functionalized dibenzhydryl substituents in α-diimine Pd catalyzed ethylene polymerization and copolymerization have been found effective and worthy to report.

There are some comments to be considered.

1: This type of work seems to be routine work by using such type of complexes. But, it contains an efficient catalytic system having good impact in polymerization reaction.

Thank you for your useful comments. We have revised this manuscript. Sterically bulky diarylmethyl-based ligands have received increasing attention in the field of late-transition-metal catalyzed olefin polymerization. Ortho-substituents may have significant impact on the performance of diarylmethyl-based α-diimine Pd catalysts in ethylene polymerization and copolymerization with methyl acrylate (MA)..

  1. In scheme 1 the ligands L3 and L4 having substituents are R3 and R4, which are changes in palladium complexes R1 and R2 its very confusing.

Thanks a lot for catching this. Scheme 1 has been revised according to your suggestions:

Scheme 1. Synthesis of the ligands and the Pd catalysts.

  1. For the thermal stability of all the complexes TGA data should be mention because palladium catalyst stability Generally, α-diimine palladium catalysts with ether auxiliary ligands for ethylene polymerization are prone to be deactivated at high temperature.

Ligand structures are crucial in determining the properties of Brookhart catalysts for ethylene polymerization. Significant advances have been made in the design and modifications of the N-aryl substituents. It is well established that bulky N,N-diaryl-substituents were required to enable high stability, high activity and high polymer molecular weights.

The branching density can be controlled by the polymerization temperature in Table 1. The produced highly branched polyethylenes with no Tm (determined using DSC), indicating that these polymers were amorphous.

Compared to the classic palladium catalyst, this ortho-functionalized dibenzhydryl substituted Pd catalysts displayed good thermal stabilities, maintaining high activity of up to 8.8 × 105 g PE (mol Pd h)−1 at higher temperature(80 oC).

  1. The morphology of the polymer should be characterized using SEM and TEM analysis.

Thanks for your useful suggestion. These α-diimine based Pd precatalysts led to the formation of branched (>69/1000 C) polyethylenes or copolymers. As such, totally amorphous polymeric materials with very poor mechanical properties were obtained. The morphology of the branched PEs or copolymers are tangled and disorganized amorphous solids, so it is meaningless to study the morphology of these polymers using SEM and TEM analysis.

Reviewer 3 Report

Report on the manuscript by Wang, Tan and co-workers submitted to Polymers.

The authors reported the synthesis and characterization of a series of new ortho-substituted α-diimine ligands and their Pd(II) complexes. All the new compounds were well characterized (multinuclear NMR, ESI-MS…), including by X-ray diffraction studies for all the metal complexes.  

The Pd pre-catalysts were applied, after activation with a borate salt, to the homo-polymerization of ethylene and co-polymerization of ethylene and methyl acrylate. The catalysts showed good activity and co-monomer incorporation up to 3 mol%.

The paper is well written, well-organized and the data support the conclusions.

The work is suitable for publication after minor corrections (see below) in Polymers.

Minor corrections:

  • Scheme 1: on the right column the substituents in Pd3 and Pd4 are R3 and R4 (similarly than for 4,5 and L3, L4). Also, in the square on the right, for 4, L3, Pd3 and 5, L4, Pd4, the substituents are R3 and R4.
  • In scheme 1 also, reaction conditions could be added in the footnote and yields could be added within the scheme.

Author Response

Reviewer: 3

The authors reported the synthesis and characterization of a series of new ortho-substituted α-diimine ligands and their Pd(II) complexes. All the new compounds were well characterized (multinuclear NMR, ESI-MS…), including by X-ray diffraction studies for all the metal complexes. The Pd pre-catalysts were applied, after activation with a borate salt, to the homo-polymerization of ethylene and co-polymerization of ethylene and methyl acrylate. The catalysts showed good activity and co-monomer incorporation up to 3 mol%. The paper is well written, well-organized and the data support the conclusions. The work is suitable for publication after minor corrections (see below) in Polymers.

Minor corrections:

(1) Scheme 1: on the right column the substituents in Pd3 and Pd4 are R3 and R4 (similarly than for 4,5 and L3, L4). Also, in the square on the right, for 4, L3, Pd3 and 5, L4, Pd4, the substituents are R3 and R4.

Thanks for your useful suggestion. Scheme 1 has been revised according to your suggestions:

Scheme 1. Synthesis of the ligands and the Pd catalysts.

(2) In scheme 1 also, reaction conditions could be added in the footnote and yields could be added within the scheme.

Thanks for your useful suggestion. The reaction conditions are very clear in Scheme 1, and yields of the compounds have been discussed in the text.
